# Antigen Presentation of mRNA-Based and Virus-Vectored SARS-CoV-2 Vaccines

**DOI:** 10.3390/vaccines9080848

**Published:** 2021-08-03

**Authors:** Ger T. Rijkers, Nynke Weterings, Andres Obregon-Henao, Michaëla Lepolder, Taru S. Dutt, Frans J. van Overveld, Marcela Henao-Tamayo

**Affiliations:** 1Science Department, University College Roosevelt, 4331 CB Middelburg, The Netherlands; n.weterings@ucr.nl (N.W.); m.lepolder@ucr.nl (M.L.); f.vanoverveld@ucr.nl (F.J.v.O.); 2Microvida Laboratory for Medical Microbiology and Immunology, St. Elizabeth Hospital, 5022 GC Tilburg, The Netherlands; 3Department of Microbiology, Immunology and Pathology, Colorado State University, Fort Collins, CO 80523, USA; Andres.Obregon@ColoState.edu (A.O.-H.); taru.Dutt@colostate.edu (T.S.D.); Marcela.Henao_Tamayo@ColoState.edu (M.H.-T.)

**Keywords:** mRNA vaccine, viral vector vaccine, Spike protein, antigen presentation, polyethylene glycol, platelet factor 4, thrombosis

## Abstract

Infection with Severe Acute Respiratory Syndrome Coronavirus 2 (SARS-CoV-2) causes Coronavirus Disease 2019 (COVID-19), which has reached pandemic proportions. A number of effective vaccines have been produced, including mRNA vaccines and viral vector vaccines, which are now being implemented on a large scale in order to control the pandemic. The mRNA vaccines are composed of viral Spike S1 protein encoding mRNA incorporated in a lipid nanoparticle and stabilized by polyethylene glycol (PEG). The mRNA vaccines are novel in many respects, including cellular uptake and the intracellular routing, processing, and secretion of the viral protein. Viral vector vaccines have incorporated DNA sequences, encoding the SARS-CoV-2 Spike protein into (attenuated) adenoviruses. The antigen presentation routes in MHC class I and class II, in relation to the induction of virus-neutralizing antibodies and cytotoxic T-lymphocytes, will be reviewed. In rare cases, mRNA vaccines induce unwanted immune mediated side effects. The mRNA-based vaccines may lead to an anaphylactic reaction. This reaction may be triggered by PEG. The intracellular routing of PEG and potential presentation in the context of CD1 will be discussed. Adenovirus vector-based vaccines have been associated with thrombocytopenic thrombosis events. The anti-platelet factor 4 antibodies found in these patients could be generated due to conformational changes of relevant epitopes presented to the immune system.

## 1. Introduction

The high morbidity and mortality rate of coronavirus disease of 2019 (COVID-19) has triggered the rapid development of vaccines against its causative agent, Severe Acute Respiratory Syndrome Coronavirus 2 (SARS-CoV-2). Vaccines are the most effective way to eliminate and control the virus [1,2]. Most of the vaccines developed for COVID-19 have shown very high levels of protection. Within one year after the outbreak of the pandemic and identification of the genomic structure of SARS-CoV-2, a number of highly effective vaccines were approved and used globally, as over 2.5 billion vaccine doses have been administered [3] (dated 25 June 2021; World Health Organization). The two major categories of SARS-CoV-2 vaccines are mRNA-based vaccines and viral vector vaccines, both targeting the Spike protein of the virus [4]. Worldwide, the most used mRNA vaccines are those of Pfizer/BioNTech (BNT162b2, brand name Comirnaty) and of Moderna (mRNA-1273, brand name COVID-19 Vaccine Moderna). The most-used adenovirus vector vaccines are the ones of Oxford/AstraZeneca (ChAdOx1 nCoV-19, brand names Vaxzevria and Covishield) and Jansen/Johnson and Johnson (Ad26.COV2.S, brand name Janssen COVID-19 Vaccine), as well as the Sputnik-V and CanSino vaccines.

Both mRNA vaccines for SARS-CoV-2 as well as viral vector based vaccines have turned out to be highly effective for protection against mild and severe COVID-19 cases. After vaccination, high titers of IgG and IgA antibodies against the Spike protein are generated which, in vitro, show a virus neutralizing capacity, and cytotoxic T cells are activated [5,6,7].

The aim of this review is to delineate the molecular pathways, outside and inside of the cell, which ultimately lead to the presentation of Spike peptides to the immune system. Both the classical antigen presentation routes via MHC class I to CD8+ T cells and via MHC class II to CD4+ T cells, as well as the antigen-presenting routes for presentation to non-conventional T cells, will be reviewed and discussed.

While SARS-CoV-2 vaccines are protecting from the severe illness and deaths due to COVID-19, after large-scale implementation, rare immune-mediated side effects became apparent. In particular, anaphylactic reactions and various thrombotic or abnormal bleeding have raised concern [8,9]. These side effects may be due to abnormal handling or presentation of the vaccine or vaccine additives to the immune system, of which the potential scenarios will be discussed.

## 2. SARS-CoV-2 Antigen Presentation

### 2.1. Presentation of SARS-CoV-2 Antigens during COVID-19

SARS-CoV-2, like the other coronaviruses (e.g., SARS and MERS), is an enveloped, positive sense, single-stranded RNA virus with a genome length of ~30 kB. The life cycle of the virus within the host consist of five steps: (1) attachment, (2) penetration, (3) replication, (4) maturation, and (5) release. Attachment occurs through the binding of a virus to host receptors, and penetration occurs through the endocytosis of membrane fusion. Once the virus enters the host cytoplasm, viral contents are released, and replication is initiated. The virus takes over the host’s protein-synthesizing mechanisms to produce viral proteins (replication), which are subsequently produced (maturation) and released [10].

Coronaviruses consist of four structural proteins: a spike (S), membrane (M), envelope (E), and nucleocapsid (N). In the mechanism of infection, the Spike protein is one of the key players [10] (Yuki et al., 2020). On a mature coronavirus, the Spike protein is present as a trimer with three receptor-binding S1 heads sitting on top of a trimeric membrane fusion S2 stalk [11]. These two functional subunits have different functions; the S1 subunit binds to the host cell receptor, and the S2 subunit is responsible for the fusion of viral and cellular membranes [10]. The SARS-CoV-2 S1 unit contains a receptor-binding domain (RBD) that specifically recognizes angiotensin-converting enzyme 2 (ACE2) as its receptor. To fuse membranes, the Spike protein needs to be proteolytically activated at the S1/S2 boundary so that SA1 dissociates and S2 undergoes a structural change [11]. Together with host-derived factors such as the cell surface serine protease transmembrane protease, serine 2 (TMPRSS2), viral uptake and cellular fusion with the host membrane is promoted [12].

Once having entered the cytoplasm of the host cell, the virus is uncoated, and viral genomic RNA is released. Translation of the two large open reading frames, ORF1a and ORF1b, of the viral RNA is initiated immediately. These reading frames encode 15–16 nonstructural proteins (nsp) which compose the viral replication and transcription complex, which includes RNA-processing and RNA-modifying enzymes and a proofreading function necessary for maintaining the coronavirus genome. ORFs that encode structural and accessory proteins are transcribed from the 3′ one third of the genome to form a set of sub genomic mRNAs (sg mRNAs) [12]. Translated structural proteins transit through the ER-to-Golgi intermediate compartment (ERGIC). Here, the interaction with N-encapsidated, newly synthesized viral RNA takes place and results in the budding of the lumen of secretory vesicular compartments. Completed virions are secreted from the infected cells by exocytosis [12].

Throughout the viral infection cycle, the infected cells present viral peptides within major histocompatibility complex (MHC) class I antigens. Class I presented viral peptides will lead to the activation of CD8+ T cells, which are capable of lysing virus-infected tissue cells [13,14]. CD8+ T cells become activated, proliferate, and differentiate into virus-specific effector and memory T cells. In the early stages of infection, professional antigen-presenting cells (dendritic cells, macrophages, and also B-lymphocytes) present viral peptides to CD4+ T cells through the MHC class II molecules (Figure 1) [14].

### 2.2. mRNA Vaccines

Nucleic acid vaccines containing antigens encoded by either DNA or RNA are delivered through the use of viral vectors (such as adenoviruses) or non-viral delivery systems (e.g., electroporation or lipid nanoparticles) [15]. These types of vaccines offer solutions for issues caused by more traditional vaccines, such as the risk of a reversion to virulence in live-attenuated vaccines or the need for additional adjuvants [16]. Nucleic acid vaccines can also be very effective, as they mimic a live, in situ infection by expressing antigens after immunization. This primes both B and T cell responses and builds an adaptive immune response directed toward the encoded target antigen [15].

Within RNA vaccines, two general classes of mRNAs are commonly used as vaccine genetic vectors: non-replicating and self-amplifying mRNA [16]. Although both utilize the host cell translational machinery for the production of the antigen target and launch of an adaptive immune response, non-replicating mRNA only encodes protein antigen(s) of interest, while self-amplifying mRNA is also capable of encoding proteins, allowing for RNA replication [17]. The current COVID-19 mRNA vaccines are non-replicating vaccines.

The mRNA vaccines against SARS-CoV-2 Spike protein were developed by Moderna and Pfizer/BioNtech in record time, with initial vaccinations occurring less than a year after this novel coronavirus was sequenced [18,19,20,21]. Even though mRNA vaccines appear to be simple (consisting of a lipid envelop surrounding mRNA molecule encoding for the protein of interest), the foundation to optimize their safety and efficacy profiles was previously established through the pioneering work of multiple individuals, in part enabling this success story. Four key technical aspects will be briefly discussed below in the context of antigen uptake and presentation.

#### 2.2.1. mRNA

The mRNA molecule in both vaccines consists of the following elements: a 5′ Cap attached to the 5′ UTR, followed by the coding sequence for the SARS-CoV-2 Spike protein, a 3′ UTR, and a long poly-A tail [22,23]. In general, elements within the mRNA were optimized to increase its stability, maximize protein translation, and reduce unwanted side effects due to innate immune activation. However, a side-by-side comparison of each company’s individual approach will be possible once currently undisclosed information becomes available upon protection of intellectual property rights. Importantly, “minor” variations in mRNA elements impacting mRNA stability, translation, and Spike protein expression efficacy could potentially explain differences in the immunogenicity profiles described for both vaccines, as discussed below. The following is a description of the methodology to optimize some mRNA elements, which is based on references cited in publications reporting initial results for SARS-CoV-2 vaccine trials.

In both vaccines, the mRNA is synthesized by in vitro transcription of a DNA fragment encoding for all elements, except the 5′ Cap. A Cap1 structure is covalently attached to the 5′ UTR either co- or post-transcriptionally via different capping enzymes used by Pfizer/BioNTech and Moderna, respectively [22,24]. Thereafter, the final product is purified using affinity chromatography on oligo-dT in order to remove the impurities generated during transcription (such as double-stranded RNA), which could potently activate the innate immune response [25]. Moderna optimized the 5′ UTR using machine learning techniques trained on ribosomal loading profiles of a reporter gene library, in which the 5′ UTRs contained completely random sequences [26]. This was further tested and validated on thousands of human 5′ UTR and variants associated with diseases in humans. BioNTech also used a library to optimize the 3′ UTR [27]. Specifically, the reporter gene was linked at the 3′ end to random cDNAs obtained by reverse transcription of fragmented mRNA isolated from human dendritic cells. Surprisingly, upon several rounds of enrichment for highly expressing constructs, the most effective 3′ UTR corresponded to a mitochondrial, non-coding rRNA. Adding a second 3′ UTR from a different gene further enhanced mRNA stability and protein expression. Finally, BioNTech optimized the poly-A tail to 120 bp in length by testing the levels of a reporter protein expressed from mRNAs differing in poly-A tail lengths [28]. This study also characterized an idoneous poly-A tail as being unmasked (i.e., no other nucleotides except adenines should be present in the 3′ end) [28].

The mRNA molecules included in the Pfizer/BioNtech and Moderna vaccines were modified by replacing N1-methyl-pseudouridine (Ψ) with uridine during in vitro transcription. Karikó et al. correctly hypothesized that unmodified nucleotide bases, such as those present in RNA transcribed in vitro, were responsible for RNA’s strong activation of the innate immune system [29]. In contrast to the ubiquitous presence of modified bases in mammalian RNA (excluding mitochondrial RNA), innate immune mechanisms evolved to detect and become activated upon encountering RNA containing unmodified bases [29]. Indeed, significant reduction in dendritic cell activation was observed when transfected with RNA transcribed in vitro in the presence of Ψ or N1-methyl-Ψ instead of uridine [30,31]. Similar results were observed in vivo in animals injected parentally with mRNA synthesized in the presence of modified bases [32,33]. Furthermore, higher and longer expression levels for the protein of interest were observed in animals injected with mRNA containing modified bases. As was recently reported, higher translation efficacy could result from stable secondary structures occurring in mRNAs containing N1-methyl-Ψ [34], as well as the greater ribosomal density associated with modified mRNA [35]. Alternatively, the presence in mRNA of Ψ or N1-methyl-Ψ could increase its stability by circumventing degradation by ubiquitous RNases, a downside to mRNA-based vaccines [30,36,37]. It should be noted, however, that some groups recently questioned the absolute requirement of modified nucleosides for efficient protein expression from mRNA-based vectors [38]. High protein expression levels were observed in mice, pigs, and non-human primates injected with mRNAs containing unmodified nucleosides. This was achieved by optimizing the 5′ and 3′ UTRs, as well as codon usage. When possible, uracil-containing codons were replaced by alternative codons encoding for the same amino acid but lacing uracil nucleosides (i.e., increasing the GC content) [38].

#### 2.2.2. Protein Sequence

BNT162b2 and mRNA-1273, which are Pfizer/BioNtech and Moderna anti-SARS-CoV-2 mRNA vaccines, respectively, both encode for the full-length Spike protein consisting of a signal sequence, S1, and S2 domains (comprising the large extracellular domain including the RBD), followed by the short transmembrane and cytoplasmic domains [19,22]. Pfizer/BioNtech additionally performed clinical trials with BNT162b1, a secreted, truncated version of the Spike protein consisting of the RBD region [21]. Despite eliciting a potent humoral and cellular immune response [21,39], BNT162b1 is not currently used, due in part to more frequent side effects [19]. Specifically, systemic events including fever, chills, fatigue or headaches were reported in older individuals (>65 years of age) vaccinated with BNT162b1 [19]. Furthermore, BNT162b2 encoding the full-length Spike protein had the added benefit of including additional epitopes potentially targeted by the immune response. Upon ribosomal translation, the signal sequence targets the protein to the ER, where it starts undergoing significant post-translational modifications. Aside from removal of the signal sequence, glycosylation (both N and O-type) as well as disulfide bond formation occur in this organelle. As it continues through the secretory pathway en route to the cell membrane, the first of two activating proteolytic events occurs: a furin-dependent cleavage not described in the related SARS-CoV-1 Spike protein [40]. To stabilize the vaccine’s pre-fusion conformation upon furin cleavage and maximize eliciting antibodies against the native viral Spike protein, two mutations for proline residues were engineered in the vaccine’s S2 subunit [22,41]. The end product in host cells expressing these mRNA vaccines is a surface-exposed, membrane-anchored, glycosylated, and trimerized Spike protein resembling the 3-D structure of the native viral Spike protein, to the extent that it interacts with its cognate receptor, hACE2 [22].

#### 2.2.3. Lipid Nano Particle (LNP)

In order for the translation of exogenous mRNA to occur in the cytoplasm of the target cell, it first has to cross the barrier imposed by the hydrophobic lipid cell membrane. Lipid nanoparticles (LNPs) are the most commonly used platforms for mRNA delivery and are mainly composed of ionizable cationic lipids, cholesterol, phospholipids (such as distearoylphosphatidylcholine), and polyethylene glycol (PEG)-lipid [42,43]. Ionizable cationic lipids participate in nanoparticle packaging by interacting with negatively charged mRNA molecules [44]. Moreover, the amine head group plays a key role in mediating endosomal uptake. Specifically, increased LNP uptake and mRNA translation was observed when LNPs included ionizable cationic lipids derivatized with amine head groups having pKa values of 6.6–6.8 [45]. Interestingly, efficient mRNA translation was shown to be associated with rab7-dependent late endosomes and their association with mTOR signaling [46]. Even though these formulations of LNPs + mRNA accumulated in late endosomes, only a small yet sufficient percentage of the vaccine’s mRNA was translocated to the cytoplasm for translation [46]. Thus, further work is required to precisely define the mechanism wherein endosomally localized mRNAs are shuttled cytoplasmically. Finally, the amine head group present in ionizable cationic lipids was closely associated with the LNP pharmacokinetics in vivo [45]. Upon administration, biodegradable LNPs that were rapidly cleared from injected tissues were less likely to induce inflammation and tissue damage [47] while, importantly, conserving adequate mRNA translation levels [45].

#### 2.2.4. Immune Response

To determine the fate of the mRNA vaccines upon in vivo administration via the intramuscular route, some studies used fluorescently labeled LNPs carrying mRNA encoding for fluorescent or luminescent reporter proteins. Brito et al. showed that, with a self-amplifying mRNA vaccine encoding green fluorescent protein, the following intramuscular immunization vaccine antigens are expressed by myocytes [48]. Similar data were obtained in a rhesus macaque model [49]. Using a previous iteration of LNPs, Moderna showed leukocyte migration to the site of LNP injection but not upon control injections with PBS. Vaccine formulations were efficiently uptaken at the site of injection by phagocytic cells such as neutrophils and different types of monocytes and macrophages; however, macrophages represented the major vaccine cell type at the draining lymph nodes. Interestingly, not all transfected leukocytes efficiently translated the target protein, and a clear dissociation between high vaccine uptake yet low protein expression was observed for neutrophils [50]. In humans, FDG-PET scans of recently vaccinated patients showed increased uptake in the deltoid muscle, corresponding to the vaccine injection site as well as in the ipsilateral (enlarged) axillary lymph nodes [50]. While latter data do not allow for differentiating between different cell types, they do indicate that intramuscular injection leads to the metabolic activation of local tissue.

Further studies evaluated how mRNA vaccines induced a potent humoral response. Upon influenza mRNA vaccination of non-human primates, germinal centers were observed in the draining lymph nodes and, importantly, antigen-specific follicular helper T (Tfh) cells were detected within these structures [51]. Collectively, this represents an ideal niche conducive to B cell activation, antibody isotype switching, and affinity maturation, leading to long-lived memory B cells and plasma cells. Indeed, a major goal of COVID-19 mRNA vaccines consists of eliciting high titers of high-affinity antibodies against Spike proteins and RBDs capable of neutralizing SARS-CoV-2 infection [18,19].

Using overlapping peptide pools covering the Spike protein, additional subsets of antigen-specific T cells were reported in humans upon vaccination with Moderna and Pfizer/BioNTech COVID-19 mRNA vaccines [24,52]. The CD4+ T cells responded to peptides from both the S1 and S2 subunits, validating the decision to use mRNA encoding the full-length Spike protein instead of a secreted RBD. Furthermore, based on cytokine profiles, the vaccine skewed helper CD4+ T cells to Th1 [24,52,53], an ideal scenario to circumvent Th2 responses responsible for vaccine-associated enhanced respiratory disease (VAERD) [54]. Intriguingly, whereas both vaccines induced helper CD4+ T cells, CD8+ T cells were only reported for the Pfizer/BioNTech vaccine [24,52]. The significance of this result remains to be determined, as both vaccines elicited comparable levels of protection (~95%) against SARS-CoV-2 strains circulating at the time clinical trials were performed. However, this could have important repercusions against emerging viral variants with higher virulence or transmissibility. Pfizer/BioNTech recently characterized in greater detail human CD8+ T cell responses in a small cohort of vaccinated individuals. Interestingly, CD8+ T cells responding to peptides from the S2 subunit were identified in some unvaccinated individuals, possibly cross-reacting to epitopes shared with seasonal coronaviruses [24]. Furthermore, some overlap was observed for epitopes recognized by CD8+ T cells upon vaccination and natural infection. Clearly, additional epitope mapping is required to understand how mRNA vaccines elicit cellular immune responses.

The mechanism of action of an mRNA vaccine is very similar to the mechanism of viral infection. By means of the translational machinery of the host cells, the mRNA is translated into proteins. These proteins may undergo post-translational modification and either function within the cell or be secreted. Proteasomes degrade cytoplasmic proteins, thus generating antigenic peptide epitopes that are transported to the ER and loaded onto MHC class I molecules (Figure 2). MHC class I can present these peptides on the surface of the cell for specific CD8+ T cells. Alternatively, the secreted exogenous proteins can be taken up by professional antigen-presenting cells, either residing in the tissue or draining lymph nodes, and then be processed and presented in MHC class II [17]. In mRNA-vaccinated individuals (BNT162b1), T-cells can be detected to be secreting interferon-γ upon in vitro restimulation with SARS-CoV-2 peptides, which confirms the induction of CD4+ Th-cells through MHC class II [39]. Professional antigen-presenting cells also can present exogenous antigens, which are processed via alternative intracellular routing and presented via MHC class I (cross-presentation) [54,55].

There are multiple advantages to using mRNA-based vaccines over traditional approaches. As mentioned before, mRNA vaccines, at the conceptual level, combine the simplicity, safety, and focused immunogenicity of subunit vaccines with the favorable immunological properties of live viral vaccines. mRNA vaccines are molecularly defined to encode only the specific antigen of interest and no other excess information. This means that in the case of a SARS-CoV-2 vaccine, the mRNA does not encode the entire virus, but only the S-protein. This greatly reduces the complications associated with biological vaccine production (such as genetic variability). An important benefit of RNA-based vaccines is the enormous flexibility of vaccine design and production. The antigen encoding sequence (the ORF) can be easily modified at specific locations or codon optimized to improve translation or engineered to guide the antigen to the desired intracellular compartments to improve antigen presentation [55]. Modifications such as point mutations, deletions, or the removal of glycosylation sites could all potentially affect antigenicity, immunogenicity, and overall vaccine efficacy. Moreover, next to additions to the coding sequence, the half-life of mRNA, the pharmacokinetics of protein expression (such as magnitude and duration), and immunogenicity are all available for fine-tuning via modifications of, for example, the 5′ and 3′ UTRs and optimization of the length of the poly-A tail [56]. The mRNA could also be tailored in such a way to provide potent adjuvant stimuli to the innate immune system by direct activation of RNA-specific receptors, which may reduce the need for additional adjuvants [36]. 

### 2.3. Adenoviral Vector Vaccines

A relatively new group of vaccines is those based on viral vectors [57]. This type of vaccine gained importance for vaccination against pathogens that did not yield sufficient immune responses in the past when approached by conventional vaccines. After the initial success of adenoviral vector-based therapeutic drugs in clinical settings [58,59,60], vaccines based on viral vectors were developed for the prophylaxis of infectious diseases such as Ebola and malaria. The choice for adenoviruses as vectors is obvious, because this group of viruses is widespread, and usually, they do not lead to serious infections and related pathology [61]. Technologically, adenoviruses are easy to grow and multiply in tissue cultures. They are thermostable and have a broad tropism, meaning they can infect a wide range of cells, and administration of these adenovirus vector-containing vaccines is easy to perform without any hurdles in the muscular and mucosal tissue. Adenoviruses are non-enveloped and have double-stranded DNA, and their medium-sized genome is about 26–48 Kbp. This size is still acceptable for easy manipulation. Adenoviruses are widespread, species-specific, have many known serotypes, and still novel serotypes are regularly described (http://hadvwg.gmu.edu; Accessed on 29 June 2021). In humans, the more than 80 serotypes are divided into seven species from A to G [62,63]. Most serotypes belong to species D.

Due to the abundant presence of adenoviruses in the human population, humoral and cellular immunity against these viruses is generally present [64,65], and initially, this was considered a drawback that needed to be solved [66,67,68]. The strategies to follow involve modification of the antigenic epitopes on the viral capsid of human Ad5. One method for this is the chemical modification by PEGylation to improve the vaccine’s efficacy via shielding or hiding of the epitopes [69]. Other useful techniques are the insertion of peptides or other types of modification of the capsid proteins. As an alternative, rare human adenoviral serotypes such as Ad26 and Ad35 [70] or chimpanzee adenoviruses can be used [66,71]. In addition, all these viruses need to be genetically modified to prevent replication when administered to humans. Normally, adenoviruses attach to membrane receptors. Ad5 will bind to the so-called coxsackievirus and adenovirus receptor CAR, which belongs to the Ig superfamily of proteins present in nearly every human cell and whose normal function is just being discovered [72]. Via clathrin-coated pits, the virus is taken into the cells, and after rupture of the endosome, viral particles are dispersed into the cytoplasm. The E1 gene is transcribed in the immediate early phase, and it is this gene which is removed to prevent viral replication when the adenovirus is used as a vector.

Replication-deficient adenovirus vectors need to be produced in cells that contain the E1 gene for their replication. Originally, human embryonic kidney (HEK293) cells, a cell line derived from a female fetus, were used as important cells for Ad5 viral vector replication [73]. Later, different cell lines with and without transfection of particular genes were also explored for other adenoviral vectors [61,74].

The working mechanism of the adenoviral vector vaccines is based on the cellular introduction of a recombinant viral genome (cDNA) containing a promotor sequence, the gene encoding the antigen, and a poly-A tail. Upon intramuscular administration, muscle cells get infected and will subsequently present a processed antigen via MHC class I and start to secrete a viral antigen to induce an immune response by activating antigen-presenting cells. The advantage of this mode of action is that both the innate and adaptive immune systems are activated, and a humoral and cellular response will result.

Currently, the vectors of choice for Sars-CoV-2 and suitable candidates for long-term protective immunity are both human and primate adenoviruses [75,76]. At the moment, there are four approved adenovector vaccines available: Ad26.CoV2.S (Janssen-Johnson & Johnson; brand name Janssen COVID-19 Vaccine), ChAdOx1 nCoV-19 or AZD1222 (Astra-Zeneca; brand names Vaxzevria and Covishield), Gam-COVID-Vac (Gamaleya National Research Centre for Epidemiology and Microbiology; brand name Sputnik-V), and Ad5-nCOV (CanSino Biologics; brand name Convidecia). All these vaccines are based on non-replicating adenoviruses, all encode full-length Spike protein, and all were proven to be efficient. Ad26.CoV2.S makes use of the human non-replicating vector Ad26 [77,78]. AZD1222 is based on the chimpanzee-derived E1-deficient adenoviral vector ChAdY25, which was demonstrated to have a low human seroprevalence [79]. Gam-COVID-Vac is a recombinant non-replicating two-vector vaccine (rAd26 and rAd5) [80], and Ad5-nCOV is based on the modified non-replicating Ad5 vector [81].

Many other viral vector vaccines are currently being developed [82], also vaccines making use of other techniques. Amongst them are vaccines from the REGA institute of the Catholic University Leuven (weakened virus, based on the yellow fever vaccine), GSK-Sanofi (adjuvanted recombinant protein-based vaccine), and Medicago (a (tobacco) plant-based virus-like particle vaccine).

## 3. Antigen Presentation of Corona Vaccine Additives

### 3.1. Unconventional Antigen Presentation Molecules

In Section 2.1, the classical routes of antigen presentation of peptide antigens in MHC class I and II for CD8+ (cytotoxic T cells) and CD4+ (T helper cells), respectively, was described. The B cell receptor (membrane immunoglobulin) can recognize and interact with either soluble or cell-bound antigens, but it does not require antigen presentation in MHC class I or II. However, the antibody response of B cells to protein antigens is dependent on T helper cells, in particular follicular helper T cells [83].

In the allergic reactions to mRNA vaccination, which occur as rare but serious adverse events, IgE antibodies to PEG have been implicated. The frequency of anaphylaxis after injection is approximately 11.1 in 1 million for the Pfizer/BioNTech vaccine [9]. The CDC in the USA reported 21 cases of anaphylaxis out of 1.8 million doses administered. For the Moderna vaccine, the CDC reported 2.5 cases per million doses administered [84]. AstraZeneca has also shown some cases of anaphylaxis. According to the European Medicines Agency, 41 out of 5 million people vaccinated with AstraZeneca showed possible signs of anaphylaxis. Jansen has also shown few hypersensitive reactions [85].

The mechanisms behind these anaphylactic reactions are still unknown; however, a hypersensitivity reaction to either polyethylene glycol (PEG) in the mRNA vaccines or to polyoxyethylene-sorbitan-20-monooleate (polysorbates 80) in the virus-vectored vaccines could be the vaccine components triggering the hypersensitive reaction.

Previous studies on vaccine-associated anaphylaxis showed that additives such as gelatin, egg protein, latex, or polysorbate 80 all can elicit hypersensitive reactions in susceptible persons [86]. Case studies have shown that patients with a known allergy for PEG also can be hypersensitive to polysorbate based on cross-reactivity between the two compounds [87].

### 3.2. Hypersensitivity and PEG Antigen Presentation

It is difficult to understand why some people have an allergic response to PEG and why others do not. Previous studies have indicated that induction of hypersensitivity does depend on the dosage, the route of administration, as well as the molecular weight of PEG used [88]. Numerous cosmetic products such as toothpaste contain PEG, and therefore, everyone is continuously exposed to PEG. Many drugs are conjugated to PEG (PEGylated) to increase their half-life and efficacy [89]. It has been shown that PEGylated drugs, such as doxil, can lead to hypersensitivity reactions, especially in the case of high molecular weight PEG administered intravenously [90].

### 3.3. Antigen Presentation of PEG

PEG, which is obviously not a peptide, would have to be presented through an alternative, non-MHC pathway. A possible pathway would be CD1 antigen presentation. CD1 is a protein (family) related to the non-polymorphic and MHC class I molecules. CD1 proteins are able to present (its own and foreign lipid antigens) to T lymphocytes. Group 1 (CD1a-c) and 2 (CD1d) CD1 proteins are expressed on the cell surface and function as antigen-presenting molecules. Group 3 (CD1e) is only expressed extracellularly and is involved with the processing and editing of lipids for presentation by the other CD1 isoforms. Although structurally related (see Figure 1), a difference between MHC class I and CD1 is that the inner surface of CD1 is covered with hydrophobic residues, and the α helices differ as well. There is a deeper antigen-binding groove in CD1 (which differs per CD1 isoform).

Very limited data are available about the potential role of BTN3A in the presentation of SARS-CoV-2 antigens to the immune system. A search on PubMed (23 July 2021) with BTN3A and SARS-CoV-2 yielded zero hits. In addition, for the presentation of SARS-CoV-2 antigens by MHC class 1 related genes A and B (MICA and MICB), scarce data are available at the moment. In a study from Brazil which has not yet been published in a peer reviewed journal, Castelli et al. analyzed 86 discordant Brazilian couples, where one partner was infected and symptomatic while the partner remained asymptomatic and seronegative despite sharing the same bedroom during the infection. The authors found only a minor impact of classical MHC class I and class II genes associated with resistance. However, individuals producing higher amounts of MICA and low amounts of MICB were more susceptible to SARS-CoV-2 infection [91]. While these are preliminary data which need to be confirmed, it is interesting from the perspective that MICB can serve as an antigen-presenting molecule for γδ T cells.

### 3.4. Thrombosis and Vaccine-Induced Thrombocytopenia (VITP)

Vaccines have some mild to moderate side effects, with some seen in the clinical trials of COVID-19 vaccine development, in which thousands of volunteers have participated, and some rare cases of anaphylaxis occurred [9]. Thrombotic side effects were initially reported for recombinant genetic vaccines and have raised concerns about the safe use and development of vaccines [1]. One of the severe events in COVID-19 infection is coagulopathy leading to various thrombotic complications and even death [92]. Similar to COVID-19/SARS-CoV-2 infection-induced immune thrombocytopenia (ITP) [93], ITP post-vaccination is a possible adverse effect. The vaccine adverse event reporting system (VAERS) has documented over 160 cases (June 2021) of thrombosis or thrombocytopenia as an adverse effect of mRNA vaccines [94]. The European Medicines Agency (EMA) has reported at least 169 cases (4 June 2021) of cerebral venous sinus thrombosis (CVST) and 53 cases of splanchnic vein thrombosis (SVT) among 34 million recipients of the ChAdOx1 nCoV-19 vaccine. The increasing numbers of rare adverse events are not surprising, as vaccination numbers are constantly increasing. However, the pathogenesis or etiology of these adverse effects is not completely clear. We therefore present some possible mechanisms of vaccine-induced ITP, partly based on abnormal antigen processing or presentation.

#### 3.4.1. Possible Mechanisms or Etiology of Vaccine-Induced ITP

##### Molecular Mimicry

Vaccines may induce ITP by several mechanisms; however, molecular mimicry has been considered the classic mechanism for vaccine-induced ITP [95]. It has been suggested that the antibodies elicited by the vaccine can potentially cross-react with the surface antigens of the platelets or megakaryocytes of the host instead of the virus [96]. These autoantibody-bound platelets or megakaryocytes undergo reticuloendothelial phagocytosis, opsonization, apoptosis, or direct lysis by cytotoxic T-cells, and all of these mechanisms leading to thrombocytopenia.

##### Molecular Mimicry between SARS-CoV-2 Spike Protein and Angiotensin-I

Another possible mechanism of vaccine-induced ITP is from molecular mimicry between viruses and human peptides [97,98,99], as there are similarities (~45%) between SARS-CoV-2 Spike glycoprotein and the human protein angiotensin-I. The data relating to SARS-CoV-2-associated diseases favor molecular mimicry between the hexa- and heptapeptide of SARS-CoV-2 Spike glycoprotein and human peptides [100].

##### Translation and Antigen Presentation by MHC Class I by Platelets

Platelets have the potential for mRNA translation and protein synthesis intracellularly [101]. Platelets also present peptides via MHC class I. If SARS-CoV-2 infects platelets, mRNA translation is possible, and subsequent synthesis of the Spike protein may arise. Alternatively, platelets might present SARS-CoV-2 Spike peptides in the context of MHC class I. These mechanisms can lead to direct lysis of platelets by cytotoxic T cells or natural killer cells, causing a decrease in the platelet count and thrombocytopenia.

##### PF4 Complex

In most of the thrombotic events reported post-vaccination, the findings were consistent with antibodies against platelet factor 4 (PF4) complex. In these cases, higher platelet-activating antibodies were identified by an enzyme-linked immunosorbent assay (ELISA). The PF4 complex is the antigen in heparin-induced thrombocytopenia (HIT), an unusual autoimmune reaction seen after administering the anticoagulant heparin. Thus, in thrombotic events, post-vaccination platelet-activating antibodies against PF4 clinically mimic heparin-induced thrombocytopenia [102]. In addition, a detailed study found that anti-PF4 antibodies form a complex with the CXCL4 platelet factor and bind to the Fcγ-receptor IIa on thrombocytes. This platelet consumption leads to thrombocytopenia or ITP [8,103] (Figure 3).

##### FcγRIIa Polymorphism

Fc receptors for IgG are expressed by many immune cell types and are involved in executing and regulating antibody-mediated immune responses. Human platelets specifically express FcγRIIa, known as CD32a, which is an activating receptor with low affinity for monomeric IgG but a high affinity for IgG-opsonized immune complexes (ICs). FcγRIIa is polymorphic and expresses two different allelotypes: FcγRIIa-H131 (Histidine), with a higher affinity for human IgG2 and IgG3, and FcγRIIa-R131 (arginine), which has a lower affinity toward ICs. This polymorphism is due to a single base substitution of adenine to guanine at nucleotide position 494 [104,105]. Thus, individuals expressing the H131 allelic form of FcγRIIa receptors on their platelets could be more susceptible to ITP via HIT, autoantibodies, or IC-mediated activation. It should be noted that the association of FcγRIIa H131 with platelet activation was found in SLE [106] but not in HIT [107].

##### Tissue Plasminogen Activator (tPA)

A serine protease found on endothelial cells, called tPA, catalyzes the plasminogen to a plasmin reaction and helps break down blood clots. The ChAdOx1 nCoV-19 vaccine contains a leader sequence for tPA to create a boosting effect on Spike protein production, tailored to induce a more robust immune response [108]. However, the ChAdOx1 nCoV-19 vaccine has been shown to induce more blood clot-related issues, as mentioned before. We hypothesize that this could be partly due to the enhanced production of tPA, which can lead to hyperfibrinolysis and cause an increase in bleeding or vascular permeability.

##### Human Proteins or Peptides Present in Vaccines

Post-vaccination ITP can also be elicited by other elements, like trace amounts of proteins from the culture media, adjuvants, preservatives, and formulation carriers. Accordingly, as reported previously, it can be presumed that antibodies against those other elements can attach to many platelets or other immune cells and trigger a cross-reaction like a natural infection can [109,110].

##### Autoimmune Disorders

In highly active or overactive autoimmune disorders, the body attacks its own tissues or cells [111]. VAERS and EMA were also reported from the case studies, and many of those with severe vaccine effects had a history of autoimmune disorders, including Type-1 diabetes, rheumatoid arthritis, and hyper- or hypothyrodism. In genetically susceptible individuals, autoinflammatory dysregulation and other autoimmune mechanisms such as epitope spreading, antigen presentation, cytokine production, polyclonal activation of B cells, and bystander activation might also contribute to the severity linked to COVID-19 vaccines [112]. According to some cases of VAERS, severe thrombocytopenia may have been induced by enhancing macrophage-mediated clearance or impaired platelet production as part of a systemic inflammatory response to COVID-19 vaccination [95,113].

## 4. Conclusions

Both mRNA-based as well as viral vector vaccines with the genetic information for the SARS-CoV-2 Spike protein have turned out to induce an efficient humoral and cellular immune response. The design of these vaccines ensures that the antigens are presented to CD4+ T cells in MHC class II and to CD8+ T cells in MHC class I. The role of unconvential T cells, and the presentation of vaccine antigens to these unconventional T cells, is not completely understood at the moment. The benefits of vaccination in preventing COVID-19 must be emphasized, which far outweigh the risks of (severe) adverse events [114]. However, in rare cases, immune mediated side effects are observed, particularly hypersensitivity reactions including anaphylaxis and the combination of thrombosis and thrombocytopenia. Delineation of the molecular mechanisms underlying these adverse effects will be required to reduce the incidence and to develop adequate testing and treatment modalities.

## Figures and Tables

**Figure 1 vaccines-09-00848-f001:**
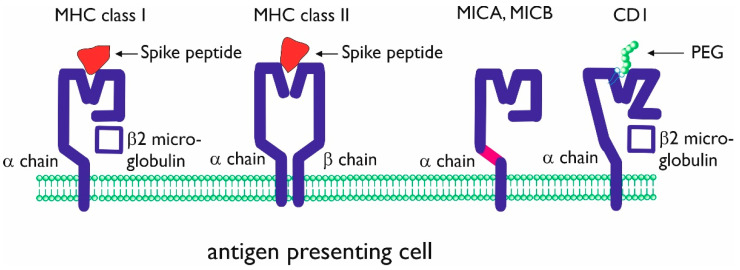
Schematic presentation of conventional and non-conventional antigen-presenting molecules major histocompatibility complex (MHC) class I and class II. The molecules are shown with (different) Spike peptides in the antigen-presenting groove. MHC class I-related molecules A and B (MICA, MICB) and CD1 are non-conventional antigen-presenting molecules consisting of a single α chain. The purple domain in the MICA, MICB α chain can be cut by metalloproteases, resulting in soluble MICA and MICB. CD1 is composed of an α chain, associated with β2 microglobulin. CD1 can present lipid antigens and (potentially) lipid-bound polyethylene glycol (PEG).

**Figure 2 vaccines-09-00848-f002:**
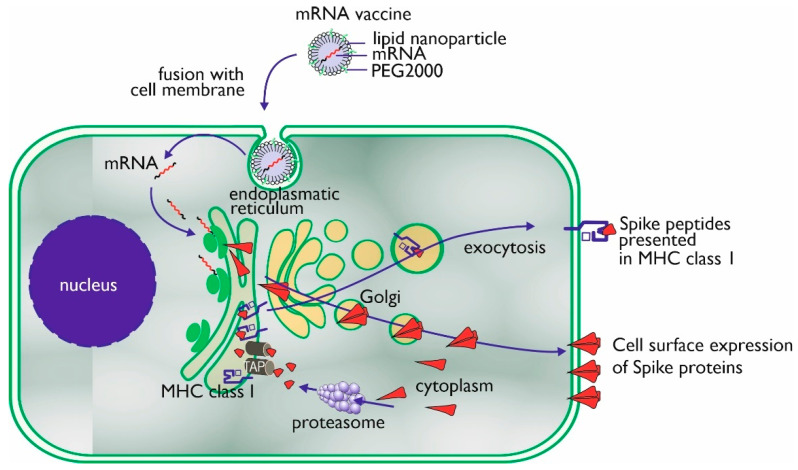
Uptake, processing, and MHC class I presentation of Spike proteins encoded by an mRNA vaccine.

**Figure 3 vaccines-09-00848-f003:**
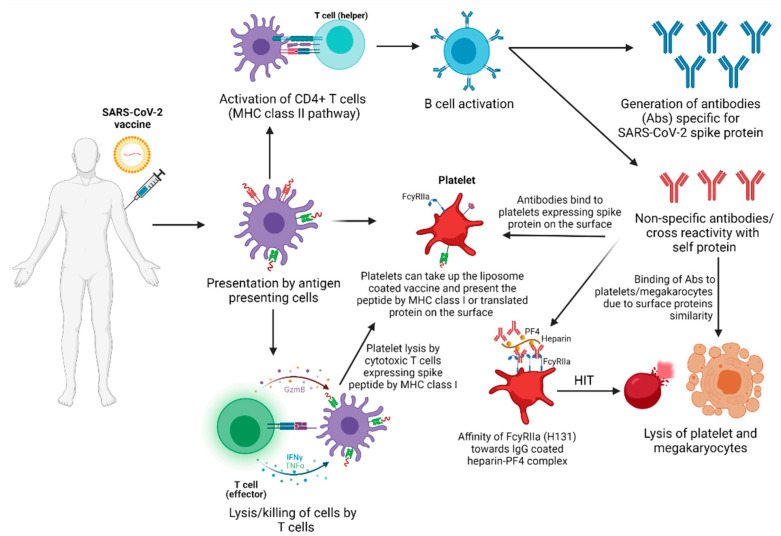
Possible etiological scenarios of vaccine-induced immune thrombocytopenia.

## Data Availability

Not applicable.

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
