# Peer review of "Antigen Presentation of mRNA-Based and Virus-Vectored SARS-CoV-2 Vaccines"

_vaccines, 2021, doi:10.3390/vaccines9080848_

Round 1

Reviewer 1 Report

Here, Rijkers et al review aspects of mRNA-based and adenovirus-vectored Covid-19 vaccines, and immune responses and side effects associated with them. Overall, the manuscript is well organized, figures are appropriate, and a large list of references is provided. One major critic, however, is that the manuscript lacks a clearly defined focus. There is little in this review on antigen presentation per se, as the viral S1 protein, produced either by a vaccine mRNA or by an infecting vector seems to be processed and presented as any other protein in its class. Vaccine design, vaccine features, formulation uptake, and immune responses dominate the review, and the title should reflect that. Second, the review is too long, especially section 3, which, although interesting, is not as much involved with the antigen presentation associated with protection but with rare side effects reported after vaccination. Not that the later are not relevant, indeed, anaphylactic and thrombotic reactions associated with Covid-19 vaccination may well deserve a dedicated review.

Comments

  1. Title. “viral vector corona vaccines” should be replaced by virus-vectored SARS-CoV-2 (or COVID-19) vaccines. The authors may want to re-consider the title of this review.
  2. Abstract. L5, “are composed of viral Spike S1…lipid nanoparticle, and stabilized…
  3. Introduction. P2, first paragraph. If the criteria is world-wide usage, likely the Sputnik V vaccine should be added to the list of vaccines here as it is used in several regions of the world.
  4. Section 2.1 “…single stranded RNA viruses with genome length of ~30 kB.”

    L6, Coronaviruses replicate solely in the cytoplasm of infected cells; no materials enter the nucleus for

    replication.

    Fig 1: the light blue domain in the MICA/MICB α chain is not well resolved in a print.

  1. Section 2.2, First line, delete “and”. First sentence in P4 “…commonly used as vaccine genetic vectors.”

    Section 2.2.1. It is not clear whether mRNAs in current vaccines are modified or not. The discussion on RNA modification is somewhat fragmented (P5, then P8). It may help the reader to consolidate this and discuss effects of RNA modification on activation of the innate immune system, efficiency of translation, and RNA stability together.

Last sentence, however Thess et al (Mol. Ther. 23,1456–1464 (2015)) obtained different results.

  1. Section 2.2.2. L4 … by short transmembrane and cytoplasmic domains. L8, please expand on the type and magnitude of side effects. L14 “… cleavage not described in the related SARS-CoV-1…”.

  1. Section 2.2.4. First paragraph, indicate that vaccine route is intramuscular. Any role of myocytes in S1 antigen presentation? Second paragraph, second sentence, “…upon influenza mRNA vaccination of non-human primates…”

  1. Section 2.3. L6 “…after release into the cytosol from endosomes, viral subparticles are terminally uncoated in nuclear pores and genomes enter the nucleus for viral transcription and replication.”

P9 third paragraph, expand on- and/or provide reference(s) for secretion of viral antigen by infected myocytes.

P9 description of individual vectored vaccines is not needed, all are based on non-replicating adenoviruses, all encode full-length S1 and all proved efficient.

  1. Section 3. It is the suggestion of this reviewer to either remove this section or reduce its size to no more than 3 pages. For example, much of section 3.1 refers to unconventional T cell numbers in Covid-19 cases with no insight on particulars of antigen presentation. Most of sections 3.2-3.4 focus on PEG antigen presentation, following the speculation that PEG-induced hypersensitivity is behind the anaphylactic reactions rarely seen after Covid-19 vaccination.

Author Response

Reviewer #1 Comments

1.1Title. “viral vector corona vaccines” should be replaced by virus-vectored SARS-CoV-2 (or COVID-19) vaccines. The authors may want to re-consider the title of this review.

We have reconsidered and agree that the suggested title indeed is better than the original one and have changed the title of our manuscript into “Antigen presentation of mRNA based and virus-vectored SARS-CoV-2 vaccines”

1.2 Abstract. L5, “are composed of viral Spike S1…lipid nanoparticle, and stabilized…

We have adapted this line of the Abstract as suggested: “The mRNA vaccines are composed of viral Spike S1 protein encoding mRNA, incorporated in a lipid nanoparticle, and stabilized by polyethylene glycol (PEG).”

1.3 Introduction. P2, first paragraph. If the criteria is world-wide usage, likely the Sputnik V vaccine should be added to the list of vaccines here as it is used in several regions of the world.

Reviewer is fully correct that we use the term “world-wide” and therefore also the other major virus-vectored SARS-CoV-2 vaccines, next to AstraZeneca and Johnson & Johnson vaccines, also the Sputnik-V and CanSino vaccines should be mentioned. We have done that in the revised manuscript.

1.4 Section 2.1 “…single stranded RNA viruses with genome length of ~30 kB.”

Of course, to avoid any confusion we have changed this line as suggested.

 1.5 L6, Coronaviruses replicate solely in the cytoplasm of infected cells; no materials enter the nucleus for replication.

We apologize for this mistake and have changed this sentence accordingly: “Once the virus enters the host cytoplasm, viral contents are released and replication is initiated”

 1.6 Fig 1: the light blue domain in the MICA/MICB α chain is not well resolved in a print.

We agree that the light blue domain in the MICA/MICB a chain didn’t stand out clearly and therefore have changed the color into purple.

1.7 Section 2.2, First line, delete “and”. First sentence in P4 “…commonly used as vaccine genetic vectors.”

We have deleted the “and” in the first line of Section 2.2. which now reads as: Nucleic acid vaccines containing antigens encoded by either DNA or RNA  are delivered through the use . . . . etc.

We have inserted “genetic” in the sentence which now reads as: Within RNA vaccines, two general classes of mRNAs are commonly used as vaccine genetic vectors . . . . etc.

1.8   Section 2.2.1. It is not clear whether mRNAs in current vaccines are modified or not. The discussion on RNA modification is somewhat fragmented (P5, then P8). It may help the reader to consolidate this and discuss effects of RNA modification on activation of the innate immune system, efficiency of translation, and RNA stability together.

We have added a phrase at the beginning of the paragraph to clarify the mRNA in current vaccines is indeed modified.

We have restructured this section to minimize fragmentation

Last sentence, however Thess et al (Mol. Ther. 23,1456–1464 (2015)) obtained different results.

We thank the reviewer for pointing out this publication, which we had missed. Indeed, Thess et al show that also sequence-engineered erythropoietin mRNA without nucleoside modification can be effective as enzyme therapy, even in animal models as large as pigs. We have added this statement to the end of Section 2.2.1: “Alternatively, the presence in mRNA of Ψ or N1-methyl-Ψ could increase its stability by circumventing degradation by ubiquitous RNases, a downside to mRNA-based vaccines [30, 36, 37]. It should be noted, however, that some groups recently questioned the absolute requirement of modified nucleosides for efficient protein expression from mRNA-based vectors [38].  High protein expression levels were observed in mice, pigs and non-human primates injected with mRNAs containing unmodified nucleosides. This was achieved by optimizing the 5’ and 3’ UTRs, as well as codon usage: when possible, uracil-containing codons were replaced by alternative codons encoding for the same amino acid but lacing uracil nucleosides (i.e., increasing GC content) [38].”

Thess A, Grund S, Mui BL, Hope MJ, Baumhof P, Fotin-Mleczek M, Schlake T. Sequence-engineered mRNA Without Chemical Nucleoside Modifications Enables an Effective Protein Therapy in Large Animals. Mol Ther. 2015 Sep;23(9):1456-64. doi: 10.1038/mt.2015.103.

1.9 Section 2.2.2. L4 … by short transmembrane and cytoplasmic domains. L8, please expand on the type and magnitude of side effects. L14 “… cleavage not described in the related SARS-CoV-1…”.

In line 4 we have modified the sentence as suggested.

Following line 8 we have added the side effects described for BNT162b1: “Specifically, systemic events including fever, chills, fatigue and/or headache were reported in older individuals (>65 years of age) vaccinated with BNT162b1 [19].”

We have modified Line 14 as suggested.

1.10 Section 2.2.4. First paragraph, indicate that vaccine route is intramuscular. Any role of myocytes in S1 antigen presentation? Second paragraph, second sentence, “…upon influenza mRNA vaccination of non-human primates…”

 In the first paragraph we have added that the route of vaccination is intramuscular.

The role of myocytes in synthesis and presentation of S1 antigen is addressed below in remark #1.12.

The second sentence of the second paragraph has been modified as suggested: “Upon influenza mRNA vaccination of non-human primates, germinal centers were observed in draining lymph nodes and, importantly, antigen-specific follicular helper T (Tfh) cells were detected within these structures”.

1.11 Section 2.3. L6 “…after release into the cytosol from endosomes, viral subparticles are terminally uncoated in nuclear pores and genomes enter the nucleus for viral transcription and replication.”

We indeed were incomplete in the description of the intracellular routing of adenovirus and have rephrased this sentence as suggested: “Via clathrin-coated pits the virus is taken up by the cells and after release into the cytosol from endosomes, viral subparticles are terminally uncoated in nuclear pores and genomes enter the nucleus for viral transcription and replication.”

1.12 P9 third paragraph, expand on- and/or provide reference(s) for secretion of viral antigen by infected myocytes.

Little is known about antigen presentation and secretion by infected myocytes. Brito et al. have shown, with a self-amplifying mRNA vaccine encoding green fluorescent protein, that following intramuscular immunization vaccine antigens are expressed by myocytes [48]. Similar data were obtained in a rhesus macaque model [49].

In humans, FDG-PET scans of recently vaccinated patients show increased uptake in the deltoid muscle, corresponding to the vaccine injection site as well as in the ipsilateral (enlarged) axillary lymph nodes [50]. While latter data do not allow to differentiate between different cell types, it does indicate that intramuscular injection leads to metabolic activation of local tissue.

We have incorporated above text into the first paragraph of section 2.2.4 and added the relevant papers to the list of references

Brito LA, Chan M, Shaw CA, Hekele A, Carsillo T, Schaefer M, Archer J, Seubert A, Otten GR, Beard CW, Dey AK, Lilja A, Valiante NM, Mason PW, Mandl CW, Barnett SW, Dormitzer PR, Ulmer JB, Singh M, O'Hagan DT, Geall AJ. A cationic nanoemulsion for the delivery of next-generation RNA vaccines. Mol Ther. 2014 Dec;22(12):2118-2129. doi: 10.1038/mt.2014.133.

Eifer M, Tau N, Alhoubani Y, Kanana N, Domachevsky L, Shams J, Keret N, Gorfine M, Eshet Y. Covid-19 mRNA Vaccination: Age and Immune Status and its Association with Axillary Lymph Node PET/CT Uptake. J Nucl Med. 2021 Apr 23:jnumed.121.262194. doi: 10.2967/jnumed.121.262194. Epub ahead of print. PMID: 33893188.

1.13 P9 description of individual vectored vaccines is not needed, all are based on non-replicating adenoviruses, all encode full-length S1 and all proved efficient.

We have revised this paragraph and reduced the description of the individual vectored vaccines to the absolute minimum.

1.14 Section 3. It is the suggestion of this reviewer to either remove this section or reduce its size to no more than 3 pages. For example, much of section 3.1 refers to unconventional T cell numbers in Covid-19 cases with no insight on particulars of antigen presentation. Most of sections 3.2-3.4 focus on PEG antigen presentation, following the speculation that PEG-induced hypersensitivity is behind the anaphylactic reactions rarely seen after Covid-19 vaccination.

We agree that Section 3 deals with immune mediated side-effects of vaccination, notably hypersensitivity reactions and thrombotic/thrombocytopenic events. The mechanisms behind both of these side effects are not totally clear but may very well be related to abnormal antigen presentation, either of the vaccine antigens itself or of additives. We also agree that this Section did become too elaborate. For the reason given above we don’t want to remove it completely, but we have trimmed it down as much as possible. By doing that, the list of references also is reduced from 148 to 115.

The total number of words has been reduced from13.617 to 10.651 and the number of pages has been reduced by 3.

Reviewer 2 Report

This review, Antigen presentation of mRNA based and viral vector corona vaccines by  summarizes Ger Rijkers and co-workers, summarizes the mode of action of vaccines against SARS-CoV2 currently implemented in Europe. This review is, to my opinion, really interesting, clear and well written.

The manuscript is a review on the current vaccines being used in Europe to fight against SARS-CoV2. Since it is a review, there are no results presented nor methodological limitations to the research and so on. To summarize this review, after explaining a bit the virus replication cycle and how antignes are presented to the immune system, the authors explain the source and the composition of the vaccines (mRNA vaccine and adeno viral vaccine) and their mode of action. Afterwards, they explain how the immune system is « educated » through these vaccines and they explain furthermore unconventional lymphocytes and antigen presentation that could be activated through these vaccines, which could explain the very rare but problematic cases of thrombosis.  

Author Response

Reviewer #2  Comments and Suggestions for Authors

This review, Antigen presentation of mRNA based and viral vector corona vaccines by  summarizes Ger Rijkers and co-workers, summarizes the mode of action of vaccines against SARS-CoV2 currently implemented in Europe. This review is, to my opinion, really interesting, clear and well written.

2.1 The manuscript is a review on the current vaccines being used in Europe to fight against SARS-CoV2. Since it is a review, there are no results presented nor methodological limitations to the research and so on. To summarize this review, after explaining a bit the virus replication cycle and how antigens are presented to the immune system, the authors explain the source and the composition of the vaccines (mRNA vaccine and adeno viral vaccine) and their mode of action. Afterwards, they explain how the immune system is « educated » through these vaccines and they explain furthermore unconventional lymphocytes and antigen presentation that could be activated through these vaccines, which could explain the very rare but problematic cases of thrombosis. 

We thank this reviewer for the compliments and kind words.

Reviewer 3 Report

This is a well written review, but unfortunately lacks focus on the actual role of the various components in the SARS-CoV-2 vaccine. I feel the focus is too general, with only a brief mention of SARS-CoV-2 at the end of the section (in most cases). 

Even where SARS-CoV-2 is mentioned, the sections still lack focus on the actual topic of the title. For example - the discussion of antigen presentation is basically a descrition of the replication cycle with a brief note on antigen presentation.

I am not sure how the Ad vector information relates to the title.

Author Response

Reviewer #3 Comments

This is a well written review, but unfortunately lacks focus on the actual role of the various components in the SARS-CoV-2 vaccine. I feel the focus is too general, with only a brief mention of SARS-CoV-2 at the end of the section (in most cases).

3.1 Even where SARS-CoV-2 is mentioned, the sections still lack focus on the actual topic of the title. For example - the discussion of antigen presentation is basically a description of the replication cycle with a brief note on antigen presentation.

We agree with this remark, which was also made by Reviewer #1. We have reduced the length of our manuscript by deleting several sections not closely related to antigen presentation. We also have restructured a number of paragraphs. By doing this, the review is more focused on the antigen presentation.

3.2 I am not sure how the Ad vector information relates to the title.

We have, as suggested by Reviewer #1, adapted the title of our manuscript into “Antigen presentation of mRNA based and virus-vectored SARS-CoV-2 vaccines”

Round 2

Reviewer 1 Report

The authors made the suggested changes and improved their manuscript.

Reviewer 3 Report

The section on the antigen presentation of SARS-CoV-2 is still just a description of the replication cycle with only a brief mention of antigen presentation.